# SIRT3 Deficiency Sensitizes Angiotensin-II-Induced Renal Fibrosis

**DOI:** 10.3390/cells9112510

**Published:** 2020-11-20

**Authors:** Xiaomeng Feng, Han Su, Xiaochen He, Jian-Xiong Chen, Heng Zeng

**Affiliations:** Department of Pharmacology and Toxicology, School of Medicine, University of Mississippi Medical Center, Jackson, MS 39216, USA; goalmesy@163.com (X.F.); suhan168302050@gmail.com (H.S.); xhe2@umc.edu (X.H.); jchen3@umc.edu (J.-X.C.)

**Keywords:** SIRT3 deficiency, hypertension, renal, fibrosis, pericyte, iron overload

## Abstract

Background: Sirtuin 3 (SIRT3) has a crucial role in the cardiovascular diseases. Our previous study revealed that SIRT3 knockout (SIRT3KO) promoted cardiac pericyte–fibroblast transition. In this study, we investigated the involvement of pericyte and iron in angiotensin II (Ang-II)-mediated renal fibrosis in the SIRT3KO mice. Methods and Results: NG2-DsRed mice and NG2-DsRed-SIRT3 knockout (SIRT3KO) mice were infused with saline or Ang-II (1000 ng/kg/min) for 4 weeks. Renal fibrosis, iron content and reactive oxygen species (ROS) were measured. Masson’s trichrome staining showed that SIRT3KO enhanced Ang-II-induced renal fibrosis. Immunostaining showed that Ang-II treatment increased the number of NG2-DsRed+ cells in the kidney, and SIRT3KO further enhanced NG2-DsRed+ cells. Moreover, SIRT3KO promoted pericyte differentiation into fibroblasts as evidenced by co-staining NG2-DsRed/FSP-1. Furthermore, DsRed/FSP-1+ and DsRed/transforming growth factor-β1 (TGF-β1)+ fibroblasts were elevated by SIRT3KO after Ang-II infusion. Ang-II-induced collagen I and TGF-β1 expression was also enhanced in the SIRT3KO mice. SIRT3KO significantly exacerbated Ang-II-induced iron accumulation. This was accompanied by an increase in acetyl-p53, HO-1 and FPN expression. Further, SIRT3KO sensitized Ang-II-induced upregulation of p47phox and gp91phox together with increased ROS formation in the kidney. Conclusion: Our study suggests that SIRT3 deficiency sensitized Ang-II-induced renal fibrosis by the mechanisms involved in promoting differentiation of pericytes into fibroblasts, exacerbating iron overload and accelerating NADPH oxidase-derived ROS formation.

## 1. Introduction

Hypertension is one of the major causes of renal diseases [1,2,3]. The mechanisms of kidney disease induced by hypertension are diverse and have not been fully understood. Hypertension leads to renal fibrosis, which might be via transforming growth factor-β1 (TGF-β1) and fibroblast-specific protein 1 (FSP-1) associated pathways [4,5]. In our previous study, we demonstrated that the production of TGF-β1 by pericytes and the pericyte–fibroblast transition contributed to renal fibrosis in the endothelial-specific prolyl hydroxylase domain protein-2 (PHD2) knockout mice [6]. Pericytes are mesenchymal cells residing around vessels. They were identified as the progenitors of fibroblasts that contribute to the deposition of extracellular matrix [7]. Recent studies have proven that pericytes play an important role in the development of hypertensive nephropathy [8]. Iron overload, which causes accelerated oxidative stress [9], plays an important role in hypertension-induced renal injury [10]. Moreover, hypertension has been shown to cause endothelial dysfunction, which also contributes to renal injury [11].

Sirtuin 3 (SIRT3), a mitochondrial nicotinamide adenine dinucleotide (NAD)+-dependent deacetylase, plays a key role in hypertension-induced renal injury [12,13]. Recent studies have shown that SIRT3 activation could alleviate the development of mouse hypertensive renal fibrosis by the suppression of endothelial-to-mesenchymal transition [12] and that the activation of SIRT3 signaling could ameliorate mouse injury induced by hypertension [13]. In our previous study, we found that the ablation of SIRT3 caused mouse coronary microvascular dysfunction and impaired cardiac recovery post myocardial ischemia [14]. Our study also showed that SIRT3 knockout (SIRT3KO) mice increased reactive oxygen species (ROS) formation and reduced capillary density in the obesity heart [15]. Our study further demonstrated that specific knockout of SIRT3 in EC resulted in mouse diastolic dysfunction and significantly increased perivascular fibrosis in the mouse coronary artery [16]. Furthermore, our study suggested that angiogenic capacities and colony formation were significantly impaired in SIRT3KO-endothelial progenitor cells (EPCs) compared to wild-type EPCs and that the loss of SIRT3 further enhanced ROS formation and apoptosis in EPCs [17]. Using NG2 pericyte tracing reporter NG2-DsRed mouse crossed with SIRT3KO mice, we demonstrated that Ang-II induced cardiac fibrosis and cardiac remodeling partly by the mechanism involving SIRT3-mediated pericyte–myofibroblast/fibroblast transition [18].

However, whether pericytes are involved in the pathogenesis of hypertensive renal fibrosis in SIRT3-deficient mice has not been studied yet. In this study, using NG2-DsRed mice to trace pericytes, we aimed to investigate the mechanism of hypertension-induced renal fibrosis involving SIRT3-associated pericyte pathway.

## 2. Methods

All procedures conformed to the Institute for Laboratory Animal Research Guide for the Care and Use of Laboratory Animals. It was also approved by the Animal Care and Use Committee of University of Mississippi Medical Center (protocol identifier: 1280C). The investigation conformed to the National Institutes of Health (NIH, Bethesda, MD, USA) Guide for the Care and Use of Laboratory Animals (NIH Pub. No. 85–23, Revised 1996).

### 2.1. Experimental Animal Model and Treatment

Wild-type (WT) control, SIRT3 knockout (SIRT3KO) (Stock#012755) mice and NG2 tracing reporter NG2-DsRedBAC (Cspg4-DsRed) (Stock#008241) mice were obtained from Jackson Laboratory (Jackson Laboratory, Bar Harbor, ME, USA) and were bred by our laboratory. SIRT3KO mice were determined by polymerase chain reaction using the following primers: SIRT3KO common, 5′-CTT CTG CGG CTC TAT ACA CAG-3′; SIRT3KO wild-type reverse, 5′-TGC AAC AAG GCT TTA TCT TCC-3′; SIRT3KO mutant reverse, 5′-TAC TGA ATA TCA GTG GGA ACG-3′. NG2-DsRedBAC mice were determined using the following primers: NG2-DsRed transgene forward, 5′-TTC CTT CGC CTT ACA AGT CC-3′; NG2-DsRed transgene reverse, 5′-GAG CCG TAC TGG AAC TGG-3′; NG2-DsRed positive control forward, 5′-CTA GGC CAC AGA ATT GAA AGA TCT-3′; NG2-DsRed positive control reverse, 5′-GTA GGT GGA AAT TCT AGC ATC C-3′ (Integrated DNA Technologies, Coralville, Iowa, USA). SIRT3KO mice were crossed with NG2-DsRedBAC mice. Among these crossed mice, homozygous SIRT3KO mice were selected by PCR firstly. Then, NG2-DsRed-SIRT3KO mice were selected from homozygous SIRT3KO mice for the experiments as previously reported [18]. Experiments were performed on male mice at 4–7 months of age.

For inducing hypertension, both SIRT3KO and control mice (9 mice/group) were infused with angiotensin (Ang)-II (1000 ng/kg/min) for 4 weeks via subcutaneously implanted Alzet mini-osmotic pumps (Durect Corporation, Cupertino, CA, USA) under anesthesia [19].

### 2.2. Histological and Immunofluorescence Analysis

The renal tissues (*n* = 3 mice/group) were fixed in neutral buffered 10% formalin solution (SF93–20; Fisher Scientific, Pittsburgh, PA, USA) and embedded in frozen optimal cutting temperature compound (4585; Fisher Health Care, Houston, TX, USA). Frozen sections and paraffin sections were prepared (8 µm in thickness). Masson’s trichrome staining (paraffin sections) was performed to measure the degree of renal fibrosis (blue). ROS (frozen sections) was measured by dihydroethidium (DHE) staining. Iron content (frozen sections) was measured by Prussian blue staining (Abcam, Cambridge, MA, USA). The area percentage of fibrosis, iron content and ROS fluorescence intensity were quantified by measuring six random microscopic fields using image-analysis software (Image J, NIH, Bethesda, MD, USA). 

NG2-DsRed+ cells were observed directly in fresh frozen sections. Immunostaining of NG2 antibody (1:100; Abcam) and DsRed antibody (1:100; Abcam) for pericytes, immunostaining of fibroblast-specific protein 1 (FSP-1) (1:100; Abcam) for fibroblasts and immunostaining of TGF-β1 primary antibodies (1:100; Santa Cruz) were also conducted in fresh frozen sections. These immune-stained sections were incubated with second antibodies conjugated with fluorescein isothiocyanate (1:500). A Nikon microscope, Nikon digital camera and Nikon software (Nikon, Japan) were used to capture images. Six random microscopic fields were analyzed using image-analysis software (Image J, NIH, Bethesda, MD, USA). 

### 2.3. Western Blot Analysis

Mouse renal cortex tissues (*n* = 3–4 mice/group) were collected and homogenized in lysis buffer. The homogenates were centrifuged at 16,000× *g* at 4 °C for 15 min. A BCA Protein Assay Kit (Pierce Co, Rockford, IL, USA) was used to analyze the protein concentrations. Equal amounts (20 µg) of the protein were separated by 10% sodium dodecyl sulfate polyacrylamide gel electrophoresis and transferred to a polyvinylidene difluoride (PVDF) membrane. The membranes were blocked with 5% nonfat dry milk in Tris-buffered saline and incubated with the following primary antibodies overnight: collagen I (1:1000; Abcam, Cambridge, MA, USA), TGF-β1 (1:1000; Santa Cruz, CA, USA), angiopoietin-1 (1:1000; Sigma-Aldrich, Saint Louis, MO, USA), angiopoietin-2 (1:1000; Santa Cruz, CA, USA), acetyl-p53 (1:500; Abcam, Cambridge, MA, USA), heme oxygenase 1 (HO-1) (1:1000; BD transduction, San Jose, CA, USA), ferroportin (1:1000; Novus Bio, Littleton, CO, USA), PFKFB3 (1:1000; Abcam, Cambridge, MA, USA), GLUT1 (1:1000; Cell Signaling, Danvers, MA, USA), p47phox (1:1000; BD transduction, San Jose, CA, USA), gp91phox (1:1000; BD transduction, San Jose, CA, USA) and β-actin (1:1000; Cell Signaling, Danvers, MA, USA). After washing, the membranes were incubated for 2 h with an anti-rabbit or anti-mouse secondary antibody coupled to horseradish peroxidase (1:5000; Santa Cruz, CA, USA). Densitometric analyses were carried out with image acquisition and analysis software (Bio-Rad).

### 2.4. Endothelial Cell Glycolysis Analysis 

Endothelial cells (ECs) were isolated from thoracic and abdominal aorta of WT and SIRT3KO mice and maintained in the EGM-2 medium (Lonza, MD, USA), as previously described [14,16,20]. Endothelial glycolysis was determined using XFe24 extracellular flux analyzer from Seahorse Bioscience, following the manufacturer’s instructions. ECs were seeded at 2.5 × 10^4^ cells per well in a 24-well Seahorse XFe24 plate (Agilent, CA, USA) one day prior to the assay. The cells were then washed twice with unbuffered XF base medium (Agilent) and incubated at 37 °C without CO_2_ for 1 h. Three initial baseline measurements were performed by using the XFe24 Seahorse analyzer followed by the injection of glucose (final concentration: 10 mM) and three additional measurements. Extracellular acidification rate (ECAR) was calculated in the Wave software (Agilent) and normalized to protein content in each well. 

### 2.5. Statistical Analysis

All analyses were performed using Statistical Package for Social Sciences version 20.0 (SPSS, Inc., Chicago, IL, USA). Data are expressed as means ± S.E.M. The assumption of normality in comparison groups was determined by normality test. The significance of differences in the means of corresponding values among groups was determined by using the one-way ANOVA. The significance of differences between two values was determined using LSD test. All the statistical tests were two-sided, and *p* values < 0.05 were considered significant (* *p* < 0.05, ** *p* < 0.01, *** *p* < 0.001, **** *p* < 0.0001).

## 3. Results

### 3.1. Deficiency of SIRT3 Enhanced Ang-II-Induced Renal Fibrosis 

Western blot analysis demonstrated that SIRT3 expression was absent in the kidneys of SIRT3KO mice. After infusion of Ang-II for 4 weeks, SIRT3 levels remained unchanged in the kidneys of wild-type (WT) mice compared to WT without Ang-II (Figure 1). Next, we examined whether Ang-II infusion led to more severe renal fibrosis in the SIRT3KO mice. As shown in Figure 2, Masson’s staining showed that knockout of SIRT3 significantly enhanced Ang-II-induced renal fibrosis (Figure 2A,B). Western blot analysis further showed that knockout of SIRT3 promoted Ang-II-induced increase in the expression of collagen I and TGF-β1 in the mouse kidneys (Figure 2C,D).

### 3.2. Deficiency of SIRT3 Promoted Ang-II-Induced Recruitment and Differentiation of Pericytes

Accumulating evidence suggests an involvement of pericyte–myofibroblast transition in renal fibrosis. Using NG2-DsRed reporter mice, we investigated the potential contribution of NG2-DsRed pericytes in Ang-II-induced renal fibrosis. As shown in Figure 3, Ang-II infusion resulted in a significant increase in the number of NG2-DsRed+ cells (red) in the WT mouse kidneys. Interestingly, Ang-II-induced NG2-DsRed+ cells were further enhanced in the kidney of SIRT3KO mouse compared to WT mice with Ang-II infusion (Figure 3A,B).

We then explored the potential mechanism by which knockout of SIRT3 led to an increase in pericytes in mouse kidneys. Angiopoietins are the key regulators of pericyte recruitment during angiogenesis. Our Western blot analysis revealed that the expression of angiopoietin-1 and angiopoietin-2 was significantly upregulated in the mouse kidneys after Ang-II infusion and this upregulation was further elevated by knockout of SIRT3 (Figure 3C,D). Immunostaining of TGF-β1 (green) revealed that Ang-II infusion resulted in a significant elevation of TGF-β1 production in the glomerulus, whereas knockout of SIRT3 significantly increased Ang-II-induced TGF-β1 production in the glomerulus. Co-staining of NG2/TGF-β1 and DsRed/TGF-β1 further showed that Ang-II infusion increased TGF-β1 production in pericytes (yellow) compared to saline infusion and knockout of SIRT3 further enhanced TGF-β1 production in pericytes (Figure 3E–G). Similarly, immunostaining of FSP-1 (green) showed that mice infused with Ang-II had a significant increase of fibroblasts in the glomerulus, and knockout of SIRT3 enhanced Ang-II-induced accumulation of fibroblasts in the glomerulus. Co-staining of NG2/FSP-1 and DsRed/FSP-1 further showed that SIRT3KO promoted Ang-II-induced differentiation of pericytes into fibroblasts (yellow) in the glomerulus (Figure 3H–J). 

### 3.3. Deficiency of SIRT3 Exacerbated Ang-II-Induced Iron Overload 

Our recent study demonstrated that knockout of SIRT3 promoted p53 acetylation, which has been shown to contribute to iron overload in mouse hearts (Su H et al. unpublished study). As shown in Figure 4, Prussian blue staining showed that Ang-II infusion increased the iron content (blue) and that was further increased in the SIRT3KO mice (Figure 4A,B). Moreover, the levels of acetyl-p53 were significantly higher in the SIRT3KO mice infused with Ang-II compared to the WT mice infused with Ang-II (Figure 4C). Western blot analysis showed that the iron-overload-associated proteins (HO-1 and ferroportin) were upregulated in the WT mice infused with Ang-II. The levels of HO-1 and ferroportin were further elevated in the SIRT3KO mice infused with Ang-II (Figure 4D,E).

### 3.4. Deficiency of SIRT3 Accelerated Ang-II-Induced Endothelial Dysfunction

Our Western blot analysis also showed that deficiency of SIRT3 resulted in a significant suppression of the endothelial glucose transporter-1 (GLUT1) and glycolytic enzyme PFKFB3 when infused with Ang-II (Figure 5A,B). This was accompanied by a significant reduction of glycolysis in ECs of SIRT3KO mice (Figure 5C). Ang-II had little effect on the levels of GLUT1 and PFKFB3 in the WT mice (Figure 5A,B). Interestingly, Ang-II infusion resulted in a significant increase in the expression of NADP oxidase subunits p47phox and gp91phox in the mouse kidneys. Ang-II further increased p47phox and gp91phox expression in SIRT3KO kidneys (Figure 5D,E). Moreover, DHE staining showed that Ang-II infusion led to a significant increase in ROS formation in the WT mouse kidneys, while knockout of SIRT3 further exacerbated ROS formation (Figure 5F).

## 4. Discussion

In the present study, we revealed that deficiency of SIRT3 deteriorated Ang-II-induced renal fibrosis. SIRT3 knockout mice exhibited more severe renal fibrosis after Ang-II infusion. Using NG2-DsRed mice and NG2-DsRed-SIRT3 knockout (SIRT3KO) mice to trace pericytes, our data further showed that these abnormalities might result from the increased numbers of pericytes, the production of TGF-β1 in pericytes and the differentiation of pericytes into fibroblasts. Interestingly, Ang-II infusion caused an iron overload in the mouse kidneys, and knockout of SIRT3 further promoted iron overload, which might also contribute to the exacerbation of renal fibrosis. Additionally, knockout of SIRT3 caused an impairment of glycolytic gene expression and accelerated NADPH oxidase-derived ROS formation after Ang-II infusion. 

Renal fibrosis is an important pathological manifestation of hypertensive nephropathy [21,22]. In the present study, knockout of SIRT3 in mice exhibited increased renal fibrosis together with an upregulation of collagen I expression. These results provide strong evidence of the involvement of SIRT3 in the development of renal fibrosis after Ang-II infusion. Pericytes are believed to function as mesenchymal cells, which reside around capillaries, precapillary arterioles, postcapillary venules and collecting venules [23,24]. Accumulating evidence suggests that the pericyte–myofibroblast transition plays a critical role in the development of tissue fibrosis and is a novel mechanism contributing to renal fibrosis in the CKD [18]. In our study, Ang-II infusion resulted in an increase in the numbers of NG2-DsRed+ pericyte. Moreover, Ang-II induced an increase in the numbers of NG2-DsRed+ pericyte in the SIRT3KO mouse kidneys. The immunostaining of NG2 and DsRed showed increased pericyte recruitment similar to that of the NG2 (pericyte) tracing reporter NG2-DsRed mice. Angiopoietins, which are secreted by pericytes, are the key regulators of recruitment of pericytes during angiogenesis [16]. In the present study, the expression of angiopoietins was increased in the WT mice infused with Ang-II, suggesting that Ang-II-induced pericyte recruitment may be mediated by upregulation of angiopoietins in the kidneys. As we expected, levels of angiopoietins were further increased in the SIRT3KO mice infused with Ang-II. These results indicated that deficiency of SIRT3 sensitized the Ang-II-induced increase of pericyte recruitments. TGF-β1 is another key mediator in the regulation of pericyte recruitment and renal fibrosis in chronic kidney diseases [25]. FSP-1, which synthesizes the extracellular matrix and collagen, is a specific marker for identification of fibroblasts. In the present study, the levels of TGF-β1 and FSP-1 were significantly increased in the glomerulus of WT mice after Ang-II infusion and were further enhanced in the SIRT3KO mice. Co-staining of NG2/TGF-β1 and DsRed/TGF-β1 indicated that SIRT3KO promoted Ang-II-induced TGF-β1 production in glomerular pericytes. By co-staining of NG2/FSP-1 and DsRed/FSP-1, our study further revealed Ang-II-induced pericyte differentiation into fibroblasts in the glomerulus and showed that SIRT3KO might result in more fibrosis in the kidneys via pericyte–fibroblast transition.

SIRT3 is a NAD+-dependent deacetylase in the mitochondria. In the mitochondria, heme is the main source of synthesized iron [26,27,28]. Excessive degradation of heme by HO-1 leads to iron overload, which causes oxidative stress. It has been reported that p53 acetylation has a critical role in iron overload [29]. SIRT3 deficiency has been shown to cause striking acetylation of p53 [30]. Ferroportin is an iron exporter and plays an essential role in the export of iron from cells to blood. Elevation of iron content results in an upregulation of ferroportin expression in the heart [31]. Prussian blue staining revealed that Ang-II infusion led to a significant increase in the iron content, which was further accentuated in the kidneys of SIRT3KO mice. This were accompanied by an increase in the expression of iron-overload-associated proteins, namely acetyl-p53, HO-1 and ferroportin. These results suggested that SIRT3KO might enhance renal fibrosis via promoting Ang-II-induced iron overload.

Endothelial dysfunction also plays a crucial role in hypertensive renal injury [11]. Recent studies revealed that SIRT3 activation could alleviate the development of mouse hypertensive renal injury by improving endothelial function [32]. In our previous studies, knockout of SIRT3 in endothelial cells exhibited a reduction in the expression of endothelial glycolytic enzyme PFKFB3 and glucose transporter GLUT1 [15,16]. Consistent with these findings, the levels of PFKFB3 and GLUT1 were reduced in the kidney of SIRT3KO mice infused with Ang-II. In contrast, the levels of NADPH oxidase (p47phox and gp91phox) and ROS formation were upregulated in SIRT3KO mice, suggesting that deficiency of SIRT3 reduced glucose metabolism and enhanced ROS formation which may result in endothelial dysfunction in the kidneys.

In summary, deficiency of SIRT3 sensitized Ang-II-induced renal fibrosis by enhancing Ang-II-induced pericyte recruitment and pericyte–fibroblast transition, by promoting Ang-II-induced iron overload and by accelerating NADPH oxidase-derived ROS formation (Figure 6).

## Figures and Tables

**Figure 1 cells-09-02510-f001:**
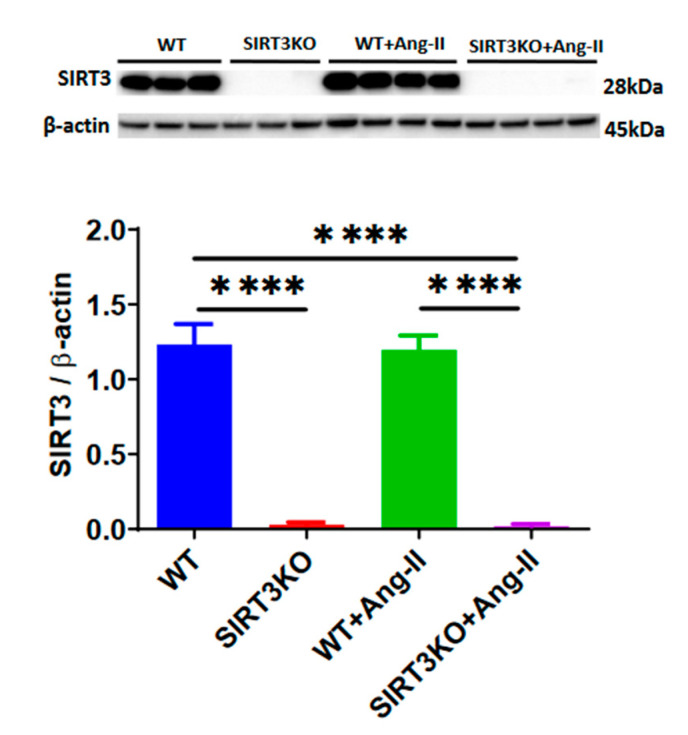
Western blot analysis showing the absence of SIRT3 expression in the kidneys of SIRT3 knockout (SIRT3KO) mice. **** *p* < 0.0001. Data are means ± S.E.M.

**Figure 2 cells-09-02510-f002:**
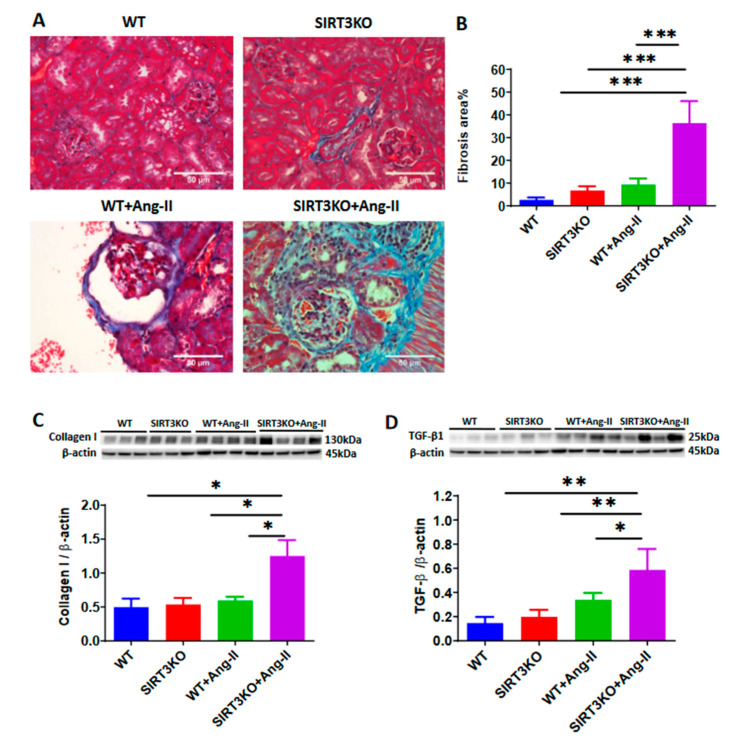
Knockout of SIRT3 enhanced Ang-II-induced renal fibrosis. (**A**,**B**) Masson’s staining showed that knockout of SIRT3 promoted Ang-II-induced renal fibrosis (blue) (*n* = 3/group). *** *p* < 0.001. Data are means ± S.E.M. (**C**,**D**) Western blots showed that knockout of SIRT3 enhanced Ang-II-induced collagen I and TGF-β1 expression in the mouse kidneys. (*n* = 3–4/group). * *p* < 0.05, ** *p* < 0.01. Data are means ± S.E.M.

**Figure 3 cells-09-02510-f003:**
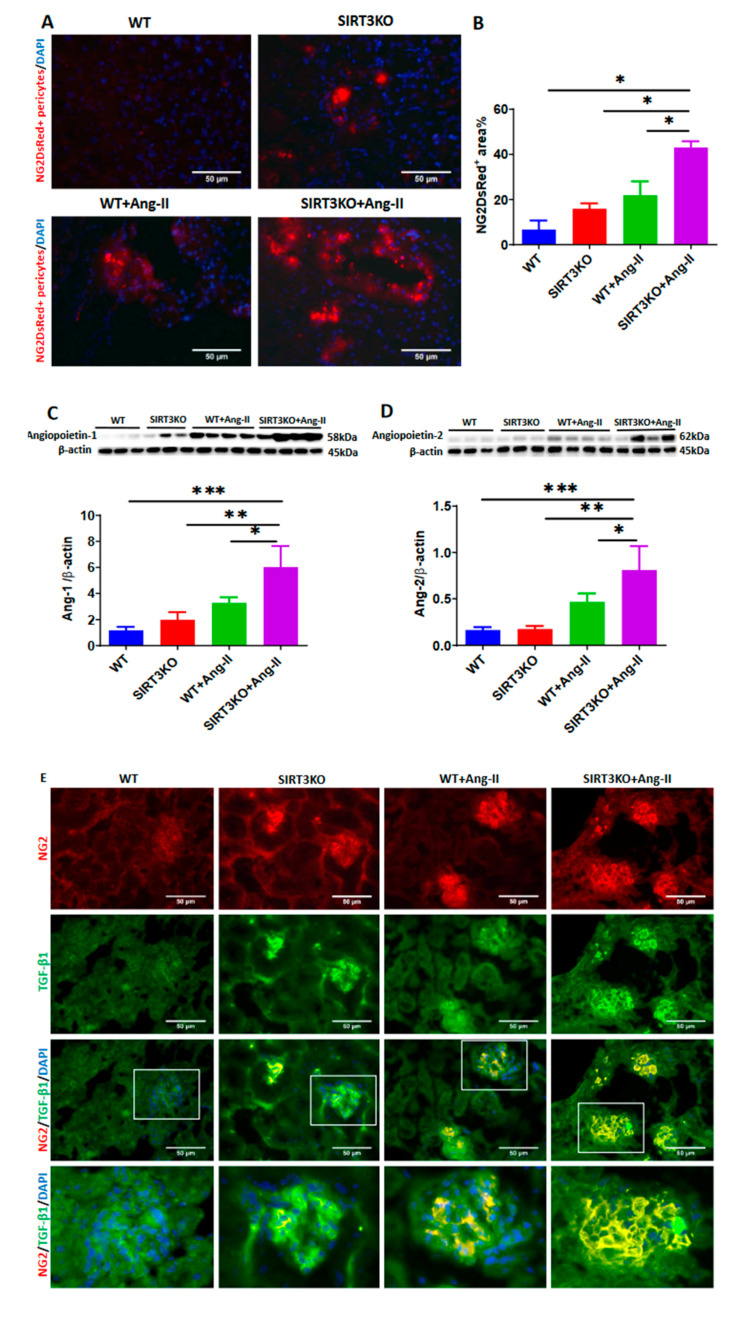
Knockout of SIRT3 promoted Ang-II-induced recruitment and differentiation of pericytes. (**A**,**B**) Ang-II infusion significantly increased the numbers of NG2-DsRed^+^ cells, while Ang-II further increased the numbers of NG2-DsRed^+^ cells in the kidneys of SIRT3KO mouse. (*n* = 3/group). * *p* < 0.05. Data are means ± S.E.M. (**C**,**D**) Western blots showing that angiopoietins were increased in the kidneys of mice infused with Ang-II and this was further elevated by SIRT3KO (*n* = 3–4/group). * *p* < 0.05, ** *p* < 0.01, *** *p* < 0.001. Data are means ± S.E.M. (**E**–**G**) Co-staining of NG2/TGF-β1 and DsRed/TGF-β1 showing that SIRT3KO promoted Ang-II-induced TGF-β1 production in pericytes (*n* = 3/group). * *p* < 0.05, ** *p* < 0.01, *** *p* < 0.001. Data are means ± S.E.M. (**H**–**J**) Co-staining of NG2/FSP-1 and DsRed/FSP-1 revealing that SIRT3KO promoted Ang-II-induced pericyte differentiation into fibroblasts (*n* = 3/group). *** *p* < 0.001, **** *p* < 0.0001. Data are means ± S.E.M.

**Figure 4 cells-09-02510-f004:**
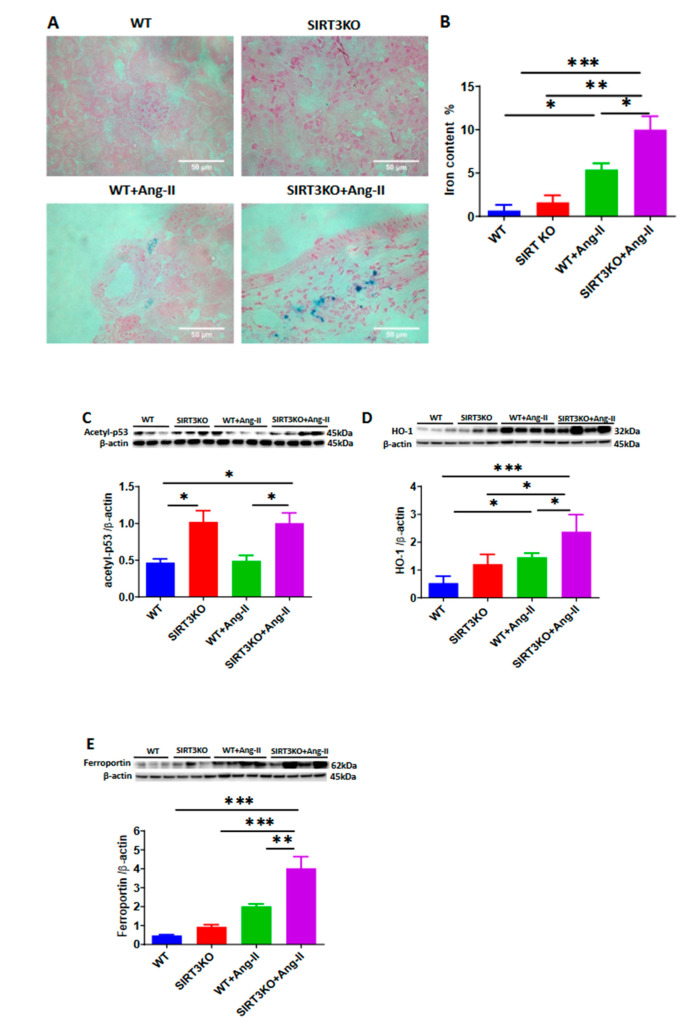
Knockout of SIRT3 exacerbated Ang-II-induced iron overload. (**A**,**B**) Prussian blue staining showed that Ang-II infusion increased the iron content in the mouse kidneys, which was accentuated by SIRT3 knockout (*n* = 3/group). * *p* < 0.05, ** *p* < 0.01, *** *p* < 0.001. Data are means ± S.E.M. (**C**–**E**) SIRT3KO increased the expression of Ang-II-induced iron overload associated proteins acetyl-p53, heme oxygenase 1 (HO-1) and ferroportin (*n* = 3–4/group). * *p* < 0.05, ** *p* < 0.01, *** *p* < 0.001. Data are means ± S.E.M.

**Figure 5 cells-09-02510-f005:**
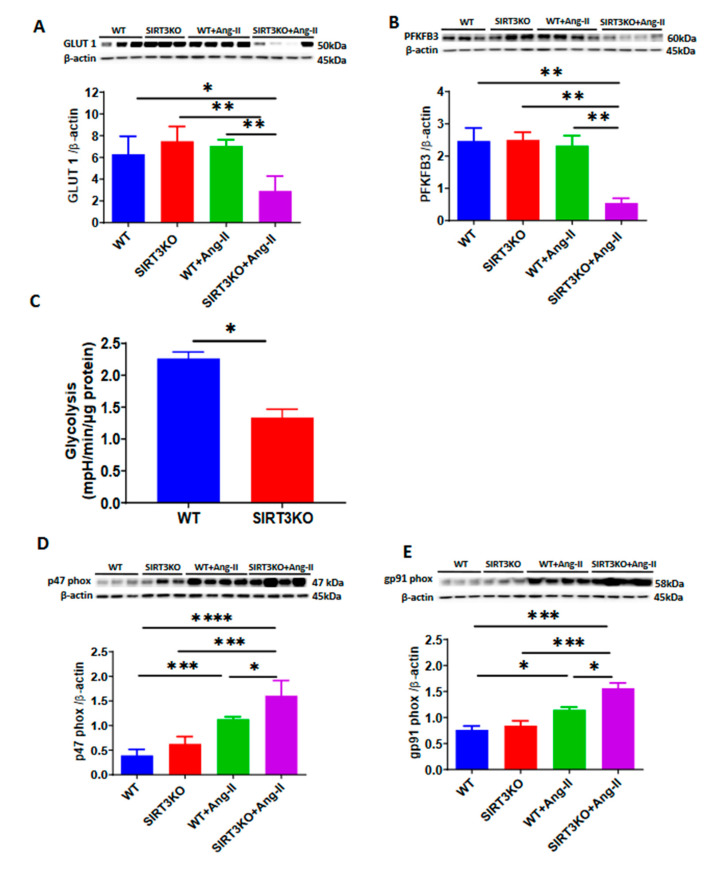
Knockout of SIRT3 accelerated angiotensin (Ang)-II-induced endothelial dysfunction. (**A**–**C**). Western blots showing that deficiency of SIRT3 sensitized Ang-II-induced reduction of endothelial glycolytic enzyme 6-phosphofructo-2-kinase/fructose-2, 6-biphosphatase 3 (PFKFB3) and glucose transporter1 (GLUT1). Endothelial glycolysis was significantly reduced in the SIRT3KO mice. (*n* = 3–4/group) * *p* < 0.05, ** *p* < 0.01. Data are means ± S.E.M. (**D**,**E**) Western blots revealing that Ang-II infusion resulted in an increase in the expression of p47^phox^ and gp91^phox^, while Ang-II further increased those in the kidneys of SIRT3KO (*n* = 3–4/group). * *p* < 0.05, *** *p* < 0.001, **** *p* < 0.0001. Data are means ± S.E.M. (**F**) Dihydroethidium staining showing that SIRT3KO enhanced Ang-II-induced reactive oxygen species (ROS) formation (*n* = 3/group). *** *p* < 0.001, **** *p* < 0.0001. Data are means ± S.E.M.

**Figure 6 cells-09-02510-f006:**
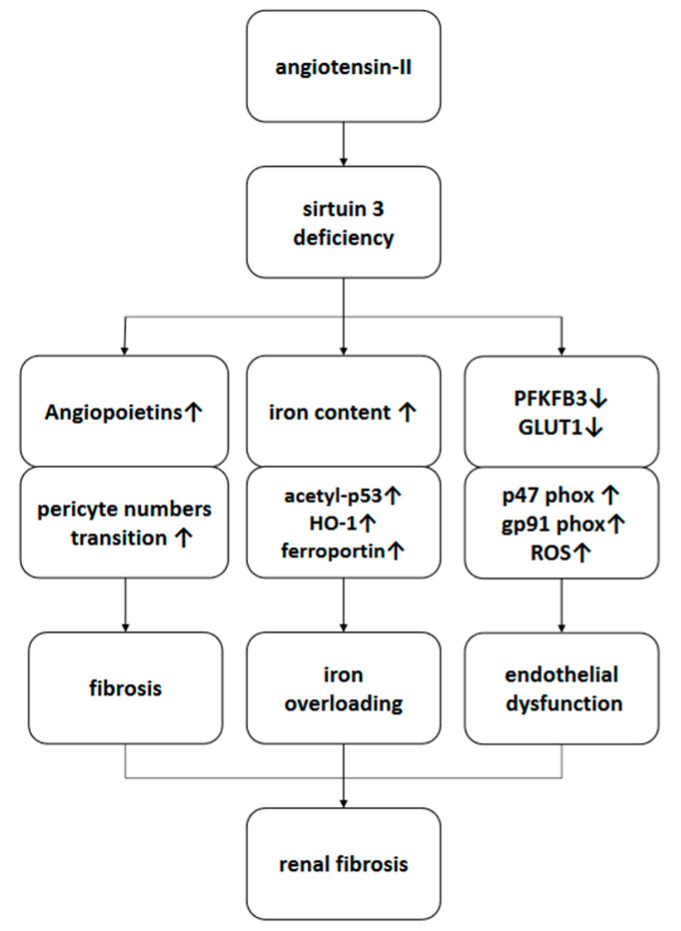
Proposed scheme of a working model of the promotion of Ang-II-induced renal fibrosis by SIRT3KO. Sirtuin 3 deficiency sensitizes angiotensin-II-induced renal fibrosis by activating differentiation of pericytes into fibroblasts, by exacerbating iron overload and by accelerating endothelial dysfunction. HO-1, heme oxygenase 1; PFKFB3, 6-phosphofructo-2-kinase/fructose-2, 6-biphosphatase 3; GLUT1, glucose transporter 1; ROS, reactive oxygen species.

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
