# Peer review of "SIRT3 Deficiency Sensitizes Angiotensin-II-Induced Renal Fibrosis"

_cells, 2020, doi:10.3390/cells9112510_

Round 1

Reviewer 1 Report

The authors show the role of Sirt3 in AngII-induced renal fibrosis by promoting pericyte differentiation into fibroblast, iron overload, and ROS formation. This paper is a well-designed experiment and suggests the critical role of Sirt3 in the renal fibrosis process. I have some questions about this experiment.

Major

  1. In the kidney, pericytes are located in the glomerulus and peritubular capillaries area. However, in figure 3, the authors show the glomerular pericytes. How about the changes in peritubular areas?

Minor

  1. What kinds of endothelial cells did you use in this experiment? In methods, authors need to clarify the isolation organ.
  2. In figure 2A, the picture quality of WT+AngII infusion group is not good. It should change to a good quality picture.
  3. How can you measure the fibrosis area? More detailed methods need.
  4. What kinds of cells are positive for Prussian blue in figure 4?
  5. How can you measure the iron content in the kidney? More detailed methods need.
  6. All presenting figures need to trimming again. Please check the figure size before publication. There is a need to delete the number of the figure.

Author Response

Reviewer 1

Major

  1. In the kidney, pericytes are located in the glomerulus and peritubular capillaries area. However, in figure 3, the authors show the glomerular pericytes. How about the changes in peritubular areas?

Thank the reviewer for this constructive comment. In the present study, we focus on NG2+ pericytes in the kidney fibrosis. Using NG2-DsRed mice to trace NG2 pericytes in the mouse kidney, we only observed a significant abundance of NG2+ pericyte in the mouse glomerular. But we did not observed a dramatic changes of NG2+ pericytes in peritubular areas.

 Minor

  1. What kinds of endothelial cells did you use in this experiment? In methods, authors need to clarify the isolation organ.

We isolated ECs from thoracic and abdominal aorta of mouse. We added this in the reversion.

  1. In figure 2A, the picture quality of WT+AngII infusion group is not good. It should change to a good quality picture.

We try our best to provide good image pictures when to publish our manuscript. These pictures are the best we could provide.

  1. How can you measure the fibrosis area? More detailed methods need.

The area percentage of fibrosis was quantified by measuring 6 random microscopic fields using image-analysis software (Image J, NIH, Bethesda, MD, USA). We added this in the reversion.

  1. What kinds of cells are positive for Prussian blue in figure 4?

In the present study, we could not identify which cells are positive for Prussian blue staining. From images of Figure 4, it looks like these cells are around renal blood vessels, so it may be fibroblast cells.

  1. How can you measure the iron content in the kidney? More detailed methods need.

The content percentage of iron staining area was quantified by measuring 6 random microscopic fields using image-analysis software (Image J, NIH, Bethesda, MD, USA). We added this in the reversion.

  1. All presenting figures need to trim again. Please check the figure size before publication. There is a need to delete the number of the figure.

We have checked all figure size and delete the number of the figure.

Reviewer 2 Report

  • In this article, Xiaomeng Feng and co-authors elegantly show in a mouse experimental model of Angiotensin II-induced fibrosis that sirtuin plays a paramount role in its pathophysiology mediated by pericyte-fibroblast transition, production of TGF-Beta1 and finally iron accumulation leading to local oxidative stress.
  • Experiments are sounded, methodologies are robust and valid.
  • Wording : please use glomerulus instead of glomerular and iron overload instead of iron overloading.
  • My main comments relate to:
  • Additional information are required about the number of mice used in each experiment: authors should give them in the methods section and also in the section results since it is not uncommon that some mice died before the end of the experiments.
  • The statistical analysis: did the authors checked the normality of the distribution (Gaussian distribution) of the different parameter analyzed allowing to perform one-way ANOVA and with which tests was normality assessed (authors have to describe it/them) ; if values are not normally distributed , then authors should use instead non-parametric ANOVA with Kruskall-Wallis test and Dunn post-tests.
  • Of note, I read in the article, in the captions of figures 2, 3, 4, 5: n= 3 or 3-4/group. If there is really such a small number of animals in each group, non-parametric statistics are mandatory

Author Response

Reviewer 2

Wording : please use glomerulus instead of glomerular and iron overload instead of iron overloading.

We thank the reviewer for point out these errors. We made these changes as the reviewer suggested.

 My main comments relate to:

  • - Additional information are required about the number of mice used in each experiment: authors should give them in the methods section and also in the section results since it is not uncommon that some mice died before the end of the experiments.

We added the number of mice used in each experiment in the methods. In our study, we do not have any animal died during the experiments. We prefer to not include animal number in the results section since these information are presented in all figures.

 The statistical analysis: did the authors checked the normality of the distribution (Gaussian distribution) of the different parameter analyzed allowing to perform one-way ANOVA and with which tests was normality assessed (authors have to describe it/them) ; if values are not normally distributed , then authors should use instead non-parametric ANOVA with Kruskall-Wallis test and Dunn post-tests. - Of note, I read in the article, in the captions of figures 2, 3, 4, 5: n= 3 or 3-4/group. If there is really such a small number of animals in each group, non-parametric statistics are mandatory.

We thank the reviewer for these comments. We have checked the normality of the distribution using normality and long-normality test. All the data are normal distribution. We included this information in the statistical analysis section.

Reviewer 3 Report

In this paper, the authors showed that although Sirt3 expression is not perturbed by angiotensin treatment, Sirt3 deficiency enhanced the renal fibrosis induced by angiotensin via collagen and TGF expression. The authors found that pericyte is increased upon angiotensin treatment and pericyte is colocalized with fibrotic markers. The authors found that Sirt3 KO increased ROS and iron content in kidney and decreased glycolysis. Several issue could be addressed for this studies.

  1. As many papers have showed Sirt3 had a role in kidney disease CKD and AKI (Marina Morigi, JCI, 2015; Wenyu Zhao, Frontiers in physiology, 2018; Yonghan Peng, Cell Death & Disease, 2019; Monica Locatelli, Scientific Reports, 2020), what is the novelty of this study to demonstrate Sirt3 KO enhances renal fibrosis?
  2. Is Sirt3 expressed in the NG2 pericyte? What is the expression level of Sirt3 in the pericyte when compared with other renal cell type? If it is too low, it is not good to delete it in those cell.
  3. What are BUN and creatinine levels in the control and disease model of WT and Sirt3 mice?
  4. In Figure 3 A and B, NG2 cell number was shown increase in WT+Ang and KO+Ang. Kidney sections have a very high autofluorescence. It is not to do quantification via IF images. FACS could be better.
  5. Did you see a mitochondrial defect difference in the models?

Author Response

Reviewer 2

  1. As many papers have showed Sirt3 had a role in kidney disease CKD and AKI (Marina Morigi, JCI, 2015; Wenyu Zhao, Frontiers in physiology, 2018; Yonghan Peng, Cell Death & Disease, 2019; Monica Locatelli, Scientific Reports, 2020), what is the novelty of this study to demonstrate Sirt3 KO enhances renal fibrosis?

We thanks the reviewer for these instructive comments. The novelty of our present study is to demonstrate the regulatory role of Sirt3 in the renal NG2 pericyte during Ang-II induced renal fibrosis by using NG2-DsRed mice to trace NG2 pericytes in the mouse kidney. Our present study also found an increased accumulation of renal iron in the SIRT3KO mice. Accumulating evidence suggested that mitochondrial iron plays a critical role in oxidative stress. This indicates that knockout of Sirt3 may regulate mitochondrial iron and ROS formation. We added these in the introduction and discussion.

  1. Is Sirt3 expressed in the NG2 pericyte? What is the expression level of Sirt3 in the pericyte when compared with other renal cell type? If it is too low, it is not good to delete it in those cell.

The reviewer raised a very important question. Sirt3 is expressed in the NG2 pericyte. So far, we did not examine the expression level of Sirt3 in renal pericyte as compared to other renal cells. These warrant to further investigation.

  1. What are BUN and creatinine levels in the control and disease model of WT and Sirt3 mice?

Unfortunately, we did not measure urine BUN and creatinine levels due to lack of equipment such as metabolic cages of mouse.

  1. In Figure 3 A and B, NG2 cell number was shown increase in WT+Ang and KO+Ang. Kidney sections have a very high autofluorescence. It is not to do quantification via IF images. FACS could be better.

We agree with the reviewer’s comments. In this study, we try to low autofluorescence as possible and keep all groups in the same. We do not the examine FACS due to lack of expertise and equipment.

  1. Did you see a mitochondrial defect difference in the models?

We did not examine the mitochondrial function in our present study. These warrant to further investigation.

Round 2

Reviewer 2 Report

This revised version has taken into account reviewers' comments and seems now suitable for publication in the journal Cells.

Reviewer 3 Report

The authors answered my questions. I don’t have any more new questions.